# Psychosocial Intervention Cultural Adaptation for Latinx Patients and Caregivers Coping with Advanced Cancer

**DOI:** 10.3390/healthcare10071243

**Published:** 2022-07-04

**Authors:** Normarie Torres-Blasco, Rosario Costas-Muñiz, Lianel Rosario, Laura Porter, Keishliany Suárez, Cristina Peña-Vargas, Yoamy Toro-Morales, Megan Shen, William Breitbart, Eida M. Castro-Figueroa

**Affiliations:** 1School of Behavioral and Brain Sciences, Ponce Health Science University, Ponce, PR 00717, USA; lrosario21@stu.psm.edu (L.R.); ksuarez21@stu.psm.edu (K.S.); cpena@psm.edu (C.P.-V.); ytoro19@stu.psm.edu (Y.T.-M.); ecastro@psm.edu (E.M.C.-F.); 2Memorial Sloan-Kettering Cancer Center, Department of Psychiatry & Behavioral Sciences, New York, NY 10065, USA; costasmr@mskcc.org (R.C.-M.); breitbartw@mskcc.org (W.B.); 3Department of Psychiatry & Behavioral Sciences, Duke University Medical Center, Durham, NC 27708, USA; laura.porter@duke.edu; 4Fred Hutchinson Cancer Center, Clinical Research Division, Seattle, WA 98109, USA; mshen2@fredhutch.org

**Keywords:** Latinx, family, meaning, communication, coping with advanced cancer

## Abstract

Latinx advanced cancer patients and caregivers are less likely to have adequate access to culturally congruent psychosocial interventions. Culturally relevant and adapted interventions are more effective within minority groups. We obtained patients’ and caregivers’ initial evaluations of the Caregivers–Patients Support to Latinx coping with advanced-cancer (CASA) protocol. A qualitative study was conducted, and an acceptance questionnaire and semi-structured interviews were conducted to culturally adapt the psychosocial intervention for Latinx coping with cancer. The semi-structured interview described and demonstrated intervention components and elicited feedback about each one. Latinx advanced cancer patients (Stage III and IV) and caregivers (n = 14 each) completed the acceptance survey, and N = 7 each completed semi-structured interviews. A total of 12 of the 14 patients and caregivers (85.7%) reported high acceptance of the goals and purposes of the intervention protocol. They also reported willingness to daily use of the content of the intervention components: Communication Skills, the Willingness of Meaning, Life has Meaning, Freedom of Will, Identity, Creative Sources of Meaning, and Homework. Most of the participants reported high acceptance (n = 9) of integrating family caregivers into therapy and the high acceptance (n = 10) of the length of the 4-session intervention.

## 1. Introduction

Adapting and developing culturally sensitive interventions is needed for Latinx families coping with advanced cancer (stage III or IV) [1,2,3]. Latinx patients coping with cancer have reported the need to include cultural values such as family and spirituality [4,5]. Family is a core value in the Latinx community and may facilitate the caring process for this group’s advanced cancer patients [6,7,8]. The inclusion of Latinx values is especially essential when evidence suggests that working with spirituality and family needs (e.g., communication) could improve psychological symptoms [6,7,8].

Including cultural values is critical to developing a culturally sensitive intervention. The adaptation of culturally sensitive interventions is more effective towards the specific culture, rather than non-adapted interventions [9]. The cultural adaptation of evidence-based interventions (e.g., meaning-centered psychotherapy and couple communication training skills) is feasible and acceptable for ethnic minority groups [9,10,11,12]. Meta-analytic evidence suggests that culturally adapted interventions targeting a specific cultural group (e.g., Puerto Ricans as part of the Latinx community) are four times more effective than those provided to groups containing various cultural backgrounds [12].

Most literature among advanced cancer patients has been developed for white patients and not adapted for Latinx patients and caregivers. Cultural adaptation of interventions designed to support advanced cancer patients and their families is a novel approach that may benefit both patients and caregivers [9,10,11,12] coping with advanced cancer. An intervention for advanced Latinx cancer patients and caregivers should consider cultural values that may affect the end-of-life process [13]. This study aimed to obtain advanced cancer patients’ and caregivers’ initial evaluations of a protocol titled: Caregivers–Patients Support to Latinx coping advanced-cancer (CASA). In this brief report, the study team included results from interviews conducted with patients and caregivers to help adjust and refine the protocol.

## 2. Method

The present study consisted of qualitative analyses for the semi-structured interview, alongside descriptive statistics for the demographic and acceptance questionnaire. The Ponce Research Institute Institutional Review Board (IRB), with the permission of the oncology clinic, approved all study procedures. Participants were recruited from an oncology clinic in southern Puerto Rico referred by the nurse or oncologist. Potential participants were introduced to the study by an IRB-approved introductory letter. This was followed by an in-person research staff meeting with participants to provide information, answer questions, and administer the Distress Thermometer to determine eligibility. The inclusion criteria for both patients and caregivers was a score > 3 on the Distress Thermometer. Those eligible and interested completed an informed consent form. A call was scheduled for those who consented to complete the questionnaire and interview. Patients and caregivers each received USD 30 for completing the interview and volunteering their time and effort. An a priori sample size of 14 participants was selected based on recommendations for qualitative studies of this nature [14,15].

The team collected demographic information and acceptance questionnaire responses from patients and caregivers separately (14 patients and 14 caregivers) and then conducted a one-time 45 min semi-structured interview (NTB) that introduced information about the intervention to participants and elicited their feedback using a standardized interview guide. Patients and caregivers were interviewed separately so their responses would not be affected. The interviewer described and provided examples of intervention components (see Table 1) and elicited feedback about each intervention component and format in the semi-structured interview. Participants were also asked about their preferred mode of intervention delivery. Interviews were audio-recorded and then transcribed. The study team reviewed the recordings and transcriptions and analyzed them using thematic content coding [14,15].

The measures used in this project are a demographic questionnaire, an acceptance questionnaire, and a semi-structured interview. The demographic questionnaire includes questions regarding biological sex, age, employment, years of school completed, marital status, insurance, annual income, type of cancer, and caregiver relationship with the patient. The acceptance questionnaire comprised 34 questions assessing the acceptance of the goals and concepts of meaning-centered psychotherapy (MCP, 16 items), couples communication skills training (CCST, 8 items), and the feasibility of the goals and therapeutic methods (10 items) of MCP and CCST. The acceptance questions assess the importance of the concepts and goals. The feasibility questions assess the likelihood of participating in a psychotherapy intervention. The semi-structured interview describes the CASA intervention content (Table 1) and includes five sections: (1) purpose and goals; (2) intervention content—MCP and CCST; (3) homework; (4) other possible topics for discussion; and (5) intervention format.

## 3. Analysis

Descriptive statistics were conducted using IBM SPSS Statistics 21 to examine the demographic and acceptance questionnaires. The semi-structured interview analyses, integration, and interpretation were completed in Spanish. A coding dictionary was developed and defined a priori, following standard deductive analysis procedures [5,6,16,17], which included developing a structured coding matrix with categories, codes, and definitions [5,6,16,17]. The qualitative codes used for the interviews included high, moderate, low, and neutral acceptance. Acceptance was noted as high when the patient or caregiver clearly stated that he or she liked the definition/question/exercise; as moderate when the patient or caregiver reported that he or she moderately liked the content or they liked it but felt that other patients or caregivers might not; and low when the patient or caregiver reported not liking the content. When the level of acceptance could not be established, it was coded as neutral.

Using the report and query functions of ATLAS.ti, the qualitative analysts (CP, LR, and KS) independently coded the transcripts and discussed divergence and convergence points [5,6,16,17]. The qualitative coders then coded the remaining transcriptions using the coding dictionary, and meetings were held to reach a consensus about the applied codes. Through consensus meetings, divergence, and convergence, points were discussed in the group until consensus was met. There were no significant differences between the coders, and if there was a difference (two to one), a consensus was reached after discussion. Reliability was conducted through team-based consensus building. The team’s previous publications include more details about the methodology [15,16,17,18]. All investigators had expertise in qualitative analysis, and the last author moderated these discussions [15,16,17,18].

## 4. Results

Demographic survey. Fourteen patient–caregiver dyads completed the demographic and acceptance questionnaire. The patients’ mean age was 59 years (SD = 11, range = 40–76), 57% were female, and 100% were Latinx. Patients’ diagnoses included stage III cancer (n = 3) and stage IV cancer (n = 10). The caregivers’ mean age was 52 years (SD = 13, range = 25–79), 57% were female, and 100% were Latinx. Among the caregivers, 9 were spouses or husbands (64.2%), 2 sisters (14.2%), 1 daughter (7.2%), 1 grandson (7.2%), and 1 friend (7.2%).

Acceptance questionnaires. Twelve patients (85.7%) and eleven caregivers (78.6%) reported high acceptance of the treatment goal. The acceptance data are summarized, and the separate voices of patients and caregivers are included in Table 2.

### 4.1. Treatment Goal

The goal of the intervention consists of exploring the meaning of life after a cancer diagnosis by sharing thoughts and feelings between cancer patients and their caregivers, facilitating a greater understanding of possible sources of meaning before and after the diagnosis, and making decisions or solving problems. On the acceptance questionnaire, patients (n = 12) and caregivers (n = 11) reported high acceptance of the intervention (see Table 2). In the qualitative interview, thirteen participants’ responses were categorized with high acceptance of the treatment goal (see Table 3). See also the illustrative quotations in Table 4.

### 4.2. Communication Skills Training

The communication skills training includes components to assist couples in communicating effectively (e.g., how to speak and listen), decreasing the avoidance of critical cancer-related issues, and supporting each other. On the acceptance questionnaire, patients (n = 13) and caregivers (n = 14) reported high acceptance of the communication skills training speaker technique (see Table 2). Moreover, patients (n = 12) and caregivers (n = 12) reported high acceptance of the listening technique in the communication skills training (see Table 2). Thirteen participants’ responses were categorized as high acceptance regarding how to listen to instructions. In the qualitative interview, all 14 participants’ responses were categorized as high acceptance of the how-to-speak instructions (Table 3).

### 4.3. Meaning Content

On the acceptance questionnaire (see Table 2), patients (n = 14) and caregivers (n = 11) reported high acceptance of the will to meaning content. The meaning content is based on the principles of Viktor Frankl’s work and his concepts of logotherapy by enhancing a sense of meaning, peace, and purpose as they approach the end of life. Patients (n = 12) and caregivers (n = 13) reported high acceptance of the identity content and experiential sources of meaning content. Patients (n = 12) and caregivers (n = 14) also reported high acceptance of the Homework: Share Your Legacy task, and patients (n = 14) and caregivers (n = 13) also reported high acceptance of the Homework: Connecting with Life task (see Table 2).

In the semi-structured interviews (see Table 3), the content identity and experiential sources of meaning were categorized as high acceptance, with 12 participants responding (see Table 3). The sources’ meaning content was categorized as high acceptance for 13 participants. The content related to freedom of will was categorized as high acceptance, with 11 responses. The content related to will meaning and life has meaning was categorized as high acceptance, with 10 responses. Regarding the meaning content of homework, connecting with life, the response of 13 participants was categorized as high acceptance. Ten responses were categorized as high acceptance regarding the Encountering Life’s Limitations and Legacy Project homework. When asked about the will to include the Legacy Project homework, 12 participants’ responses were categorized as high acceptance. Regarding the Connecting with Life homework, 11 participants’ responses were categorized as high acceptance.

### 4.4. Intervention Format

On the acceptance questionnaire, patients (n = 12) and caregivers (n = 12) reported high acceptance when asked about family integration into the therapy sessions (see Table 2). When asked about session intervention length, in the semi-structured interviews, 10 participants’ responses were categorized as high acceptance, two as low acceptance, and two as neutral (Table 3).

### 4.5. Integration of the Findings to Protocol

The text was independently reviewed, followed by consensus meetings to discuss every intervention session, provide feedback, and discuss further modifications until a consensus was reached. The most accepted content of Caregivers-Patients Support to Latinx coping advanced-cancer (CASA) was included [17]. The cultural expert (NTB) and collaborators (CP, LR, and KS) conducted the integration of the qualitative findings to develop the CASA fixed in Table 5. The most commonly endorsed content of the CASA intervention was Communication Skills: Speaker, followed by Communication Skill: Listen, Creative Sources of Meaning, and Homework: Connecting with Life. The treatment goal, the content related to identity, and the experiential sources of meaning were also accepted, as were freedom of will, the will to meaning, life has meaning, Encountering Life’s Limitations, and the Legacy project.

Dr. Rosario Costas Muñiz, the treatment and cultural expert, reviewed the text to ensure the adaptation considers the dimensions of the ecological validity model, a framework used to create culturally sensitive interventions for Hispanics [18]. Fidelity to CASA’s concepts, goals, and a theoretical model was preserved during the adaptation process to ensure language, metaphor, strategy, cultural context, and value acceptance. See Table 5 to see the intervention protocol’s content and adaptation.

## 5. Discussion

This study aimed to obtain patient-caregiver dyads’ initial evaluation of the Caregivers-Patients Support to Latinx coping advanced-cancer (CASA) protocol, specifically tailored to address spirituality and communication among Latinx patients. Included in this brief report were results from the interviews conducted with patients and caregivers to help adjust and refine the protocol. Based on the findings of the acceptance questionnaire and the semi-structured interviews, the study team refined the intervention content and format.

Latinx patients and caregivers described the communication skills content as highly acceptable and relevant to coping with patient and caregiver needs. The most endorsed content of the CASA intervention was Communication Skills: Speaker, followed by Communication Skill: Listen. These findings are consistent with the literature [6,7] suggesting the acceptance of incorporating communication skills training in patients-caregivers coping with cancer. The team only includes the most acceptable content for the adaptation of CASA. These include incorporating the content endorsed by the caregivers and patients through the semi-structured interview related to Creative Sources of Meaning, identity, and experiential Sources of Meaning. Participants also reported high acceptance of freedom of will, followed by the Will to Meaning, Life has a Meaning and Homework Encountering life limitations and Legacy project. This acceptable content was similar to the team’s previous research in adapting individual meaning-centered psychotherapy (IMCP) for Latinxs [6]. We also integrate the adapted protocol’s most accepted intervention homework by patients and caregivers: the Legacy project and Connecting with Life.

Concerning the format of CASA, many participants preferred sessions accompanied by caregivers; however, some patient–caregiver pairs reported the barrier of attending simultaneously. Thus, the intervention length was refined based on the findings of the semi-structured interview and recommendations of the cultural expert. As suggested in the findings from the acceptance survey, the format of the CASA intervention will need to match the needs and resources of patients and caregivers (e.g., videoconferences, telephone intervention, or home visits). As evidenced by the present study, most participants reported high acceptance of incorporating caregivers into the intervention and high acceptance of the intervention length.

## 6. Conclusions

In conclusion, patient–caregiver dyads found CASA’s communication and meaning content acceptable, as evidenced in the acceptance survey and semi-structured interviews. Additionally, caregivers and patients expressed the acceptance of participating in this intervention together. The results make the CASA adaptation an essential step towards refinement and piloting with Latinx families coping with advanced cancer. Furthermore, including communication skills and meaning-centered content will assist Latinx patients and caregivers with their palliative and end-of-life decision-making.

Implementing CASA in different communities is feasible because of this study’s variety of patients and types of cancers. However, cultural differences should be considered. This study may contribute to the development of cultural adaptations by proving sensitive ways to conduct interventions. Given the heterogeneity of Latinx culture, it will provide a practical way to facilitate the way interventions are conducted with Latinx patients. This line of research’s future direction should include the adapted intervention’s pilot test.

## 7. Limitation

Consenting with patients and caregivers and collecting data remotely were challenging and time-consuming. An additional limitation was that the team did not include a pilot test of the intervention. We also did not have sexual orientation and sexual identity in the demographic data. By not collecting this critical information, we could not divide the responses by gender or sexual identity or consider the gender of the interviewer for the results of this study. For future studies, we should consider the participants’ sexual orientation and sexual identity to report findings. Nonetheless, these preliminary findings suggest that a patient–caregiver intervention is acceptable and may be a promising approach to managing spirituality and communication in patients and caregivers coping with advanced cancer.

## Figures and Tables

**Table 1 healthcare-10-01243-t001:** Caregivers–Patients Support to Latinx coping advanced-cancer” (CASA) content presented in the interview. The X indicate the inclusion of intervention related content.

Content	MCP	CCST	Definition
Treatment Goal	X	X	Exploring the meaning of life after a cancer diagnosis by sharing thoughts and feelings between the cancer patient and their caregivers and facilitating a greater understanding of possible sources of meaning before and after the diagnosis and making decisions or solving problems
Communication Skill: Speaker		X	Guidelines for sharing thoughts and feelings
Communication Skill: Listen		X	Guidelines for listening to others’ thoughts and feelings
The will to Meaning	X		The need to find meaning in our existence is a basic primary motivating force shaping human behavior.
Freedom of will	X		We have the “freedom” to find meaning in our existence and to choose our attitude toward suffering.
Life has meaning	X		Frankl believed that life has meaning and never ceases to have meaning, or the potential for meaning, from the first moments of life up to the end.
Homework: Encountering Life’s Limitations	X		Three question exercise regarding encountering life’s limitations.
Identity	X		Our identity is significantly influenced by the people, roles, and other aspects of our life that give our lives meaning.
Experiential Sources of Meaning	X		Connecting w/life through love, relationships, beauty, nature, and humor.
Creative Sources of Meaning	X		Actively engaging in life via work, deeds, accomplishments/via courage, commitment, responsibility
Homework: Share Your Legacy~Tell Your Story and Legacy Project	X		Tell your story to loved ones in your life in any manner that is comfortable to you.
Homework: Connecting with Life	X		List three ways in which you “connect with life” and feel most alive through the experiential sources of love, humor, and beauty.
Four sessions		X	Content presented in four sessions
Family integration		X	The integration of family into therapy

**Table 2 healthcare-10-01243-t002:** Descriptive statistics for the acceptance questionnaire.

Content	N Patients	% Patients	N Caregivers	% Caregivers
Treatment Goal	12	85.7%	11	78.6%
Communication Skill: Speaker	13	92.9%	14	100%
Communication Skill: Listen	12	85.%	12	85.7%
The Will to Meaning	14	100%	11	78.6%
Identity	12	85.7%	13	92.9%
Experiential Sources of Meaning	12	85.7%	13	92.9%
Homework: Share Your Legacy~Tell Your Story and Legacy Project	12	85.7%	14	100%
Homework: Connecting with Life	14	100%	13	92.9%
Family integration	12	85.7%	12	85.7%

**Table 3 healthcare-10-01243-t003:** High, moderate, low, and neutral acceptance of the intervention.

Categories	Themes	Response
High acceptance of intervention content	Treatment Goal	12
	Communication Skill: Speaker	14
	Communication Skill: Listen	13
	The will to meaning	10
	Life has a meaning	10
	Freedom of will	11
	Homework: Encountering Life’s Limitations	10
	Identity	12
	Experiential	12
	Creative sources of meaning	13
	Homework: Share Your Legacy~Tell Your Story and Legacy Project	10
	Homework: Connecting with life	13
	4 sessions	10
Moderate acceptance of of intervention content	Homework: Encountering life’s limitations	2
	Identity	1
	Experiential	1
	Creative sources of Meaning	1
	Homework: Share Your Legacy~Tell Your Story and Legacy Project	2
Low acceptance of intervention content	Treatment Goal	2
	The will to Meaning	1
	Life has meaning	1
	Freedom of will	1
	Homework: Encountering Life’s Limitations	1
	Experiential	1
	Homework: Share Your Legacy~Tell Your Story and Legacy Project	2
	4 sessions	2
Neutral	Communication Skill: Listen	1
	The will to Meaning	1
	Life has meaning	3
	Freedom of will	2
	Communication Skill: Listen	1
	Homework: Encountering Life’s Limitations	1
	Identity	1
	Homework: Connecting with Life	1
	4 sessions	2

**Table 4 healthcare-10-01243-t004:** Caregivers–Patients Support to Latinx coping advanced-cancer (CASA) illustrative quotations for the high acceptance themes.

Themes	Illustrative Quotations for High Acceptance
Treatment Goal	Regarding the intervention’s treatment goal, caregiver #3 expressed: “I think, I think it is really, really good. Since, well, how do you say it? Because it helps.”
Communication Skill: Speaker	Regarding communication skills as a speaker sharing their thoughts and emotions, caregiver #5 expressed: “Well, how can I say it? Yes, they are important. They are good because one must learn to communicate, learn to express what one feels.”
Communication Skill: Listen	Regarding communication skills as a listener when the speaker is communicating, caregiver #12 expressed: “Yes, it seems fine to me. It is a way to follow some steps that can lead you to not interrupt that person who is perhaps expressing themselves at that moment, saying how they feel and letting you go through these steps. Then you can say: ‘let me hold on and let the person finish, even if I have any questions.’ Because, of course, they can be speaking, the person may be speaking. And if I interrupted them, interrupted them for a moment, I stopped the thought process they were having.”
The will to meaning, life has meaning and freedom of will	Regarding the need to find meaning in their life, patient #13 expressed: “I liked it… I liked two, the will to meaning (desire for meaning) and freedom of will (free will) because I believe that there are decisions or attitudes towards life, in the face of this situation that happened to me, as well as other people. And I think that you must face it, as you say, on the battlefield where you must fight every day.” Regarding Frankl’s approach to life and meaning, patient #14 expressed: “That life has a meaning was very important, now that we went back.’ People say, ‘one comes here just to suffer,’, but it always has meaning.” Regarding the freedom to find meaning in our lives and chosen attitudes toward suffering, caregiver #12 expressed: “That sounds interesting because, if you are telling me, to put a situation here, a family member dies, my father died, or someone… Well, I have freedom of will (free will) to choose how I feel in that situation. That’s what I’m understanding from what you are telling me. I find that reflection curious, that this person had, that any person, every person, has freedom of will (free will) to choose how they are going to feel, how they should face that situation. Regardless as I understand it, any person has that right or has the freedom of will (free will), but it’s not easy and not everyone can achieve it.”
Identity	Regarding identity, its influences, and its relationship to the meaning of life, patient #2 expressed: “I like it, because obviously one’s feelings on how one is, how one was and how one is and was after the… (referring to diagnosis).”
Experiential and creative sources of Meaning	Regarding the ways a person experiences and connects with life through love, beauty, nature, humor, or relationships, patient #14 expressed: “Yes, because not everyone has the same problem or, no matter what the problem is, not everyone can play the same way and anything you can contribute to help someone or, at least, visualize that you have…but a problem and how you can resolve it and how they can endure everything. I bet that it is welcomed.” Regarding how a person actively engages in life through work, deeds, and accomplishments, caregiver #5 expressed: “Very good, they are interesting.”
Homework: Share Your Legacy~Tell Your Story and Legacy Project	Regarding an assignment where the participant creates a project that integrates meaning, identity, and creativity to generate a sense of meaning in light of their life and diagnosis, caregiver #12 expressed: “It’s a good thing and in the end, the person thinks about this and has the desire to think: ‘Before I leave, I want to leave with a grand finale (gold medal). I want to travel, if I want to travel; I want to eat all the ice cream in the ice cream shop.’ Whatever it may be. ‘Let’s do this that I have never done before and always wanted, let’s do it.’”
Homework: Connecting with Life	Regarding an assignment where the participant lists how they connect with life through love, beauty, and humor, patient #14 expressed: “All of them lead you to be what you are as a human being. With one you are more open, and others are less open, but all of them lead us there.”
Family integration	Regarding the integration of a family member into the intervention, caregiver #4 and patient #10 expressed: “As long as the person is in agreement, yes.” Caregiver # 4 “Yes, because sometimes it is important to include another person. Yes.” Patient #10
Four sessions	Regarding the acceptance of an intervention with four sessions, patient #1 expressed: There would be four, because there is time to develop the topics.

**Table 5 healthcare-10-01243-t005:** CASA components that were culturally and linguistically adapted and integrated after the interviews and feedback. The X indicate the inclusion of intervention related content and the cultural adaptation.

Content	Included	Culturally Adapted	Adaptation
Treatment Goal	X	X	We include an introduction section to discuss the treatment goal and learn about cancer experience.
Communication Skill: Speaker	X	X	We include the CCST guidelines for sharing thoughts and feelings in session one
Communication Skill: Listen	X	X	We include the CCST guidelines for listening to others’ thoughts and feelings in session one
The will to Meaning	X	X	We include meaning-centered content in session three.
Identity	X	X	We include identity content in session two.
Experiential Sources of Meaning	X	X	We include experiential sources of meaning content in session three.
Homework: Share Your Legacy~Tell Your Story and Legacy Project	X	X	We include the Legacy project on session two homework
Homework: Connecting with Life	X		We include the Legacy project on session three homework
4 sessions		X	Content change to be presented in 4 sessions
Family integration	X	X	We integrate family related content to the intervention

## Data Availability

Not applicable.

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
