# Peer review of "Psychosocial Intervention Cultural Adaptation for Latinx Patients and Caregivers Coping with Advanced Cancer"

_healthcare, 2022, doi:10.3390/healthcare10071243_

Round 1

Reviewer 1 Report

1.      Adding more background information regarding culturally sensitive interventions is necessary. In addition, it may help your reader be more engaged with your article.

2.      Do you obtain the permission of the Oncology Clinic to recruit the participant?

3.      How long did the transcription take you compared with the original interview?

4.      Did you have any follow-ups after your original interviews? And how do you implement the follow-ups?

5.      How long did you recruit the participants?

6.      In the discussion, what is the implementation of CASA? Can it be able to implement in different cancer communities?

7.      What is the contribution of this study to the relevant literature? Please explain from a theoretical and practical perspective.

8.      The reference does not meet the format requirements of the Healthcare journal.

Author Response

Thank you for your feedback and comments. Attached the point-by-point response. 

Reviewer 2 Report

Dear authors

Congratulations on your article. Thank you for the chance of reading it.

Here are some comments:

- The introduction should clearly defend the importance of specific intervention if advanced cancer populations and specially in latinx patients and caregivers;

- You should clearly state your definition of "advanced cancer patient". 

- The title of the article is not suitable to the aim of the study;

- Was the study approved by an ethical comittee?

- Why were the participants paid to complete the interview? Do you think this might have influenced the interview?

- How would this intervention integrate with palliative care?

Best regards.

Author Response

(The authors gave the same response as above.)

Reviewer 3 Report

This is a novel project and seeks to identify components which are culturally appropriate in this supportive intervention. 

Abstract needs additional clarity around the utility of CASA. The rationale is not fully clear at the outset. However, the background evidence for this manuscript is clearly summarised in paragraph 2 of the introduction. 

Methods appropriate and well written.

Analysis is not clear. The themes titles and sub theme descriptions are confusing.  E.g., theme titles are almost synonymous with component information and following text. These terms should hold significant meaning and almost stand-alone without narration (to an extent). Also, the tables are overloaded with text (and bullet points), not well presented. 

Discussion short but again summarised well. 

Minor improvements in sentence structure e.g., introduction, line 5 - 'especially...'. 

Author Response

(The authors gave the same response as above.)

Reviewer 4 Report

This paper focuses on the need for cultural adaptation of an intervention aimed at improving the coping with cancer. It is a qualitative study but it is poorly analyzed and the results are not clearly stated.

Major concern:

1.  There were 14 patient-caregivers’ dyads that completed the interview. Was it that both patentis and caregivers participated in the interview and why is not the separate voices from patients and caregivers represented in the paper? The analysis of the qualitative data is poorly presented.

2. There is an extensive difference to have a paper presented in Science as you state in the reference list and a paper in Science Journal of Education. This indicates a very careless attitude to scientific writing. Is the rest of the manuscript also writing in such manner?

Minor comments:

1.  Please describe the interview process in more details. Was the patient and the caregivers interviewed separately or together? What was the reasons for your choice.

2. Please describe what you did in the analysis and how large the differences were between the coders. It is not sufficient to refer to your previous papers. It should be possible for another researcher do redo the study.

3. Were the interviewers and those analysing the data the same persons. Does the gender of the interviewer have an importance for the results of your study? Were some of you also the active physician for the patients?

4. Is it appropriate to present the percentage on different questions in a qualitative study? It is more than just demographics that is used quantitatively. The way you treat your data is more like a mixed-methods approach.

5. In Table 2 you write VT without explaining what that stands for.

6. The Table 2 the rating categories seem to be the important elements and the excerpts are used to illustrate. What was the results of the qualitative analysis?

7. The integration of the findings to the protocol is using the CASA components, which is not described in detail. Moreover, the Ecological Validity Model is not described and no further reference is given.

8. The legend to Table 3 says that it gives the cultural and linguistic adapted and integrated, but there is not much different in comparison to Table 1. Please adjust accordingly.

9. The reference list is not using a consistent system of presenting the detailed information necessary for the interested readers. Differences now are seen with regard how to write names of journals; year of publication; and no web links are given. I do not know if this is required, but these could assist in finding the publications easy.

Author Response

(The authors gave the same response as above.)

Round 2

Reviewer 2 Report

Nothing to add.

Congratulations on the improvement made.

Author Response

Thank you!

Reviewer 3 Report

Thank you for improving the article however, I am still unclear about the analysis. This needs to be simplified and the thematic analysis needs further thought. 

Author Response

Thank you, attached the point by point response.

Reviewer 4 Report

The manuscript has been Mich omprovad.

Author Response

Thank you! Attached the point by point response. 
